# Mechanisms of Warm-Water Intrusions onto the West Spitsbergen Shelf during Winter

Lukas Frank<sup>1,2</sup>, Jon Albretsen<sup>3</sup>, Ragnheid Skogseth<sup>1</sup>, Frank Nilsen<sup>1,2</sup>, and Marius O. Jonassen<sup>1,2</sup>

**Correspondence:** Lukas Frank (lukasf@unis.no)

**Abstract.** The West Spitsbergen Current, flowing northward along the continental slope in the eastern Fram Strait, represents a key pathway for warm Atlantic Water entering the Arctic Ocean. However, along the west coast of Syalbard, parts of this Atlantic Water frequently diverge from its core, intruding eastward onto the West Spitsbergen Shelf and further towards the adjacent fjords. Here, the associated excess heat has a substantial impact on the regional hydrography, as well as on the regional marine biosphere and cryosphere. This study uses a high-resolution, fully dynamical regional ocean model to investigate the mechanisms driving such warm-water intrusions during winter. Our results show that warming events on the West Spitsbergen Shelf are associated with a variety of cross-shelf exchange processes, including surface Ekman transport, upwelling of Atlantic Water from deeper slope regions across the shelf break, and topographical steering of Atlantic Water onto the shelf along shallower isobaths. The eastward displacement of the West Spitsbergen Current core itself is most frequently involved in triggering shelf warming events. Regardless of the specific mechanism, the intrusion depth on the shelf is governed by the relative density difference between the intruding Atlantic Water and the ambient shelf water. As winter progresses, increased shelf density due to cooling, brine rejection, and previous Atlantic Water intrusions, enhances the likelihood of near-surface intrusions, in contrast to intrusions typically penetrating the shelf at depth in early winter. These findings highlight the complexity and seasonality of the cross-shelf intrusions of Atlantic Water from the West Spitsbergen Current onto the West Spitsbergen Shelf. They emphasize the need for high-resolution modeling and complementing observations to fully capture the dynamics of these intrusions and their broader implications for Atlantic Water heat transport and ecosystem variability in the Svalbard region.

#### 1 Introduction

The Fram Strait (FS), located between Greenland and Svalbard, is the only deep-water connection between the Arctic Ocean and the world oceans, thus playing a crucial role for heat transport into and out of the central Arctic Ocean (Schauer et al., 2004) and subsequently for the currently observed global-scale sea ice decline (Carmack et al., 2015; Smedsrud et al., 2008; Stroeve et al., 2025). While cold water gets exported from the central Arctic with the East Greenland Current in the western part of the FS, relatively warm and saline Atlantic water (AW) is transported northward in its eastern part by the Norwegian Atlantic Front Current (NwAFC) flowing northwards along the Knipovich Ridge. Furthermore, the West Spitsbergen Current (WSC), which originates in the Barents Sea Opening where the Norwegian Atlantic Slope Current (NwASC) divides into the

<sup>&</sup>lt;sup>1</sup>The University Centre in Svalbard, Longyearbyen, Norway

<sup>&</sup>lt;sup>2</sup>Geophysical Institute, University of Bergen, Bergen, Norway

<sup>&</sup>lt;sup>3</sup>Institute of Marine Research, Bergen, Norway

North Cape Current (NCC, flowing into the Barents Sea), and the WSC (following the continental slope northwards toward Spitsbergen) transports AW northward along the West Spitsbergen Shelf (WSS, see Figure 1, Hanzlick (1983); Aagaard et al. (1987); Schauer et al. (2004); Beszczynska-Möller et al. (2012)). The NwAFC and the WSC merge west of Svalbard around 78° N (Walczowski et al., 2005). Parts of the AW recirculate west- and southwards as the Return Atlantic Current, and contribute to the Atlantic Meridional Overturning Circulation (AMOC, Bourke et al. (1988); Teigen et al. (2011); Hattermann et al. (2016)) and the deep water formation in the Greenland Sea (Rudels, 2019). Over the continental slope, typically two separate branches of the WSC can be found. The western branch is located over the deeper parts of the slope and has a strong baroclinic component (e.g. Manley et al., 1992; Schauer et al., 2004; Nilsen and Nilsen, 2007; Fer et al., 2023). Close to the WSS break, the largely barotropic eastern branch of the WSC is topographically steered over the continental slope as the northernmost extension of the NwASC, following streamlines of f/H (Coriolis parameter over water depth, Aagaard et al. (1987); Nøst and Isachsen (2003); Schauer et al. (2004); Nilsen and Nilsen (2007); Teigen et al. (2010); Nilsen et al. (2016, 2021)).

Large amounts of heat are lost from the AW on its way north, due to heat loss to the atmosphere (Boyd and D'Asaro, 1994; Walczowski and Piechura, 2011) and interaction with regionally and locally formed cold water masses (Saloranta and Haugan, 2004). This mainly occurs when AW from the WSC penetrates onto the WSS and circulates the troughs that indent the shelf. Here it encounters colder and fresher Arctic-type and coastal water masses that are transported northward along the west Spitsbergen coast on the WSS by the Spitsbergen Polar Current (SPC, Helland-Hansen and Nansen (1909); Svendsen et al. (2002); Nilsen et al. (2016); Tverberg et al. (2019); Skogseth et al. (2020); Nilsen et al. (2021)), a continuation of the coastal current originating in Storfjorden and the East Spitsbergen Current (ESC, Loeng (1991); Svendsen et al. (2002); Skogseth et al. (2005); Challet et al. (2025)) from the northwestern Barents Sea (see Figure 1). The oceanic heat transport associated with the AW intrusions onto the WSS and potentially into the adjacent fjords along the west coast of Spitsbergen has substantial implications not only for the shelf and fjord hydrography (Nilsen et al., 2008; Promińska et al., 2017; Tverberg et al., 2019; Skogseth et al., 2020; Bloshkina et al., 2021; Strzelewicz et al., 2022; De Rovere et al., 2024), but for all components of the regional climate system. Ocean heat released into the atmosphere substantially contributes to shaping the relatively mild climate compared to other regions located globally at a similar latitude (Walczowski and Piechura, 2011). Furthermore, the excess heat that is not lost to the atmosphere on the shallow shelf areas and inside the West Spitsbergen fjords prevents the formation of sea ice during winter (Cottier et al., 2007; Onarheim et al., 2014; Muckenhuber et al., 2016; Tverberg et al., 2019; Skogseth et al., 2020). Episodes of warm water intrusions into the innermost fjord arms have been shown to strongly correlate with melting and calving rates of marine terminating glaciers (Luckman et al., 2015; Torsvik et al., 2019; Foss et al., 2024). In the end, the prevalence of AW in these regions has led to habitat expansions for boreal species such as the blue mussel (Berge et al., 2005; Leopold et al., 2019), specific types of phytoplankton (Hegseth and Tverberg, 2013; Hegseth et al., 2019; Supraha et al., 2022) and zooplankton (Gluchowska et al., 2016; Hop et al., 2019) as well as fish species such as Atlantic cod and haddock (Renaud et al., 2012; Fossheim et al., 2015), which in turn affects marine mammals higher up in the food web (Descamps et al., 2017). All these factors underline the strong impact of warm AW on the regional climate and marine ecosystem on the WSS and inside the adjacent fjords. Therefore, an improved understanding is needed of the processes leading to intrusions of this water from the WSC onto the WSS in the first place.

**Figure 1.** Main ocean current systems in the Fram Strait and the western Barents Sea, based on data from Vihtakari et al. (2019). Relatively warm and saline Atlantic-type currents are shown in red, relatively cold and fresh Arctic-type currents in blue. Current abbreviations: EGC: East Greenland Current, RAC: Return Atlantic Current, WSC: West Spitsbergen Current, NwAFC: Norwegian Atlantic Front Current, NwASC: Norwegian Atlantic Slope Current, NCC: North Cape Current, ESC: East Spitsbergen Current, SPC: Spitsbergen Polar Current. The ocean bathymetry shown here and in all following maps in this study is based on the IBCAO Version 4.1 (Jakobsson et al., 2020).

Several different mechanisms have been proposed as potential drivers for the cross-shelf water mass exchange enabling the intrusion of AW onto the WSS. Persistent northerly winds along the western coastline of Spitsbergen lead to offshore Ekman transport in the surface, balanced by upwelling of AW along the shelf break and lifting deeper AW onto the WSS (Cottier et al., 2005, 2007). In a similar manner, persistent southerly winds set up direct onshelf Ekman transport in the surface layer, thus pushing AW in shallower depths onto the WSS (Goszczko et al., 2018). Furthermore, Tverberg and Nøst (2009) describe how Ekman transport combined with barotropic and baroclinic instabilities along the shelf-ward edge of the WSC (formerly explained by Teigen et al. (2010) and Teigen et al. (2011)) lead to a residual eddy overturning across the shelf break. They also show how the structure of the initial density front between the slope and shelf determines the direction of the eddy overturning (inflow in the surface or at depth). Ultimately, using a purely barotropic shelf circulation model, Nilsen et al. (2016) demonstrate how an acceleration of the WSC over the slope and/or a widening of its jet-shaped cross section leads to topographically guided

flow of AW onto the WSS and into the troughs towards the West Spitsbergen fjords. They refer to these meandering onshelf AW pathways as the Spitsbergen Trough Current (STC) and have argued that due to an increase in winter cyclones in the FS, investigated from an atmospheric perspective by Zahn et al. (2018) and Wickström et al. (2020), and correspondingly more prevailing southerly winds with subsequent Ekman transport and increased sea surface tilt, the STC became stronger in recent years.

The sheer number of different mechanisms proposed by previous studies underscores the complexity of the matter. Especially when considering the different proposed effects that wind forcing has on the cross-shelf water mass exchange, it can be expected that in reality several of the processes outlined above take place at the same time, and an eventual warming event on the WSS is the consequence of an interplay between them. However, previous investigations based on observations and/or idealized model simulations typically focus on only one process at a time. This study aims to address this gap of knowledge by analyzing WSS warming events and their connection to different forcing mechanisms based on a fully dynamical regional ocean model data set. This allows us to identify potential synergy effects between different identified drivers and attribute warming events not only to one individual, but also to combinations of these forcing mechanisms.

In the following, Section 2 describes the model setup and our data processing routines and analysis methods. Section 3 starts with an overview of all WSS warming events identified during the course of the study period. Afterwards, we present individual warming events that can be clearly assigned to one specific cross-shelf exchange process. We illustrate how certain warming events can also be triggered by combinations of these different established forcing mechanisms. Furthermore, we analyze the role that the initial shelf stratification plays for the intrusion depth, when the warm AW penetrates onto the shelf. Throughout this section, we discuss our results and put them into context with previous studies. In the end (Section 4), we summarize our findings and give an outlook on future work on the topic.

#### 90 2 Data and Methods

100

75

### 2.1 ROMS Model Setup

The numerical simulations for this study were performed using the Regional Ocean Model System (ROMS), version 3.5 (Shchepetkin and McWilliams, 2005), with the same configuration as the operational NorKyst-800 model along the Norwegian mainland coast developed in collaboration between the Norwegian Meteorological Institute (MET Norway) and the Institute of Marine Research (IMR) (Asplin et al., 2020). In particular, the model utilizes a bathymetry-following vertical coordinate system with 35 levels and was set up with a horizontal resolution of 500 m, covering the entire Svalbard region (see Figure 2 (a)). Lateral boundary conditions and tidal forcing were provided by MET Norway's operational ocean forecasting model Barents2.5 (Röhrs et al., 2023), while atmospheric forcing was obtained from MET Norway's operational AROME-Arctic numerical weather prediction model, which has a horizontal resolution of 2.5 km (Müller et al., 2017). The ROMS setup also incorporated daily freshwater runoff based on simulations with the CryoGrid land surface model, forced in turn by AROME-Arctic (Westermann et al., 2023; Schmidt et al., 2023). A simple sea-ice component, following Budgell (2005), was included to account for sea-ice effects. The total simulation covers the period from April 2019 to October 2024, but only the three

**Figure 2.** (a) Simulated SST over the whole ROMS model domain on 01.02.2020 12 UTC. (b) Zoom into the WSS region west of Spitsbergen. The red line marks the WSS break and the blue lines mark the 15 zonal sections used to analyze the warming events and the corresponding cross-shelf exchange mechanisms. The light green shading highlights the area over which the atmospheric wind forcing is evaluated. The thin black bathymetry contours have a 100 m-resolution.

winter seasons November – April 2019/20, 2020/21 and 2021/22 are considered in the present analysis, as changes in MET Norway's operational Barents2.5 forecasting system during winter 2022/23 led to discontinuities in the hydrographic boundary conditions. Raw model output consisting of hourly hydrography and current fields was averaged to a daily resolution. A brief validation of the simulated hydrography and currents against Conductivity-Temperature-Depth (CTD), Acoustic-Doppler-Current-Profiler (ADCP) and oceanographic mooring observations from the University Centre in Svalbard's (UNIS) long-term ocean monitoring program is provided in Appendix B. The validation shows a good agreement between the model simulations and the observed density and current structures, accurately capturing key circulation features on the West Spitsbergen Shelf. Despite some temperature and salinity biases — such as overly cold and fresh surface waters and slightly too fresh Atlantic Water — these errors largely compensate in density. In particular, the model realistically reproduces current variability and topographic steering on the WSS.


## 2.2 Data Processing and Analysis Methods







In a first post-processing step, the ROMS model output was horizontally re-gridded from its native curvilinear grid to a latitudelongitude coordinate system with a meridional resolution of 0.01° and a zonal resolution of 0.04°, both corresponding to approximately 1 km. The re-gridding process also ensured the proper rotation of the current components from the model grid to zonal and meridional directions. In the vertical dimension, the model output, originally defined in bathymetry-following sigma levels, was interpolated onto a regular depth grid with 10 m equidistant spacing. For the main analysis of this study, we concentrate on the WSS between Bellsund in the south and the Isfjorden Trough in the north (see the map in Figure 2 (b)). Data from 15 zonal cross-shelf sections spanning across the WSS break and slope between 77.35° N and 78.05° N were extracted from the data set (see Figure 2 (b)). A local coordinate system was defined in a way, so that each section is centered with x = 10 km at the shelf break, which was manually identified as the point of maximum curvature in the bathymetry (corresponding depths ranging between 240 m - 450 m), and extends 30 km in each direction. The southern sections were limited to  $x_{max}$ < 30 km eastward to avoid influences from deeper waters in the trough that leads to Bellsund. In the following, the positive x-range is referred to as the "shelf" or the WSS, while negative x-values correspond to the "slope". Since the angle between the WSS break and true north is small (see Figure 2 (b)), the difference between the northward current component and the actual along-slope current speed of the WSC is small. Therefore, we chose not to define individual local coordinate systems aligned with the shelf break at each individual latitude, which substantially simplifies the analysis of the meridional propagation of warming events presented in Section 3.2. We also limit the study period to the winter months, spanning from 1. November through April. During this time, the impact of AW warming events on the WSS is greatest, e.g. in the context of local sea ice formation.

A brief spectral analysis (not shown) revealed that the fortnightly tide was the only remaining tidal constituent present in the daily averaged data, and this was subsequently removed using a harmonic analysis. From the potential temperature and practical salinity fields of the model, conservative temperature, absolute salinity, in-situ density, potential density anomaly referenced to 0 dbar and heat content with reference temperature of 0°C were calculated using the Gibbs Sea Water (GSW) TEOS-10 standard (McDougall and Barker, 2011).

With a temporal resolution of 1 day, the ROMS model data time series constitute variability over a wide frequency range, from short-term inertial oscillations to the annual cycle. Since the major warming events of interest in this study occur over characteristic time scales of several days to weeks (see Section 1), it is necessary to separate the corresponding low-frequency variability from higher-frequency fluctuations, particularly those caused by transient eddies and internal waves. To achieve this, we applied a fourth-order Butterworth low-pass filter with a cutoff frequency of 20 days to the ROMS hydrographic and current time series. Due to its flat frequency response in the passband, this filter ensures minimal distortion of the retained signal. Figure A1 in the Appendix illustrates the effects of this filtering on example time series of conservative temperature, absolute salinity and current speed.

Afterwards, warming events were identified based on the frequency-filtered conservative temperature data set. We used a pseudo-Lagrangian approach, which, in contrast to the classical Eulerian view on the data (analyzing e.g. the temperature time

series at a fixed point in time, such as a mooring location), allows for a wide range of initial and boundary conditions during periods identified as warming events. At each time step and latitude section, a series of temperature contour elements were defined by  $0.2^{\circ}$ C-increment-isotherms covering the whole temperature range of the data set ( $-1.6^{\circ}$ C  $-8.2^{\circ}$ C, see Figure 3 (a)). In case of multiple, separate contour elements corresponding to the same isothermal present at the same section at the same time step, the largest one originating over the slope (westernmost point of the contour at negative x) and penetrating onto the shelf (easternmost point of the contour at positive x) was chosen for the further processing steps. For all contour elements, the position of their easternmost point was tracked independently at each latitude section throughout the study period. A warming event could then be identified as an increase in the x-locations of the tracked positions (see Figure 3 (b)) of more than 10 isotherms at all 15 zonal cross-shelf sections. Smaller events (based on only a few contour levels, not detected at an extended latitude-range etc.) were manually discarded for the further analysis. In the end, the heat content increase at any position on the shelf was calculated as the temperature difference between the isotherms located at that position before and at the end of the warming, multiplied by the mean density (1028 kgm $^{-3}$ ) and the specific heat capacity of sea water (3985 Jkg $^{-1}$ K $^{-1}$ ). The resulting data set, which serves as the basis for all further analyses presented in this study, can be found at https://doi.org/10.5281/zenodo.15188605 (Frank and Albretsen, 2025).






For the analysis of the role of the regional current systems in forcing WSS warming events, core positions of individual current branches were identified as local maxima in the northward current cross-section at each latitude and time step. An example cross-section is given in Figure 4. Minor peaks with a maximum northward speed of less than 0.05 ms<sup>-1</sup> or a prominence of less than 20% of their maximum speed were discarded (grey markers in Figure 4). Based on contour lines with a spacing of 0.02 m.s<sup>-1</sup>, the lowest contour solely contributing to each of the remaining peaks, was identified (the colored contours in Figure 4 (a)). The water column spanning over the same x-range as this contour was chosen as the corresponding current branch cross section for each branch (indicated by the dashed lines in Figure 4 (a)), and the average position of all contours at levels higher than this contour (colored in Figure 4 (b)) as the respective branch position.

Regional wind forcing was quantified as the spatially averaged surface wind stress over an area spanning from 77.3° N to 78.1° N (see the light green shading in Figure 2 (b)), calculated from the same AROME-Arctic wind data used to force the ROMS simulations. In the first place, the wind stress was calculated according to

$$\tau = \rho_a C_D W S_{10}^2, \tag{1}$$

where  $\rho_a$  is the air density (1.3gkg<sup>-1</sup>),  $WS_{10}$  is the wind speed in 10m height and  $C_D$  is the drag coefficient, which was parameterized in a nonlinear form based on Large and Pond (1981) and adapted for low wind speeds according to Trenberth et al. (1990) (see also Goszczko et al. (2018)):

$$C_D = 0.001 \cdot \begin{cases} 2.18 & \text{for } WS_{10} \le 1ms^{-1} \\ 0.62 + \frac{1.56}{WS_{10}} & \text{for } 1ms^{-1} < WS_{10} \le 3ms^{-1} \\ 1.14 & \text{for } 3ms^{-1} < WS_{10} \le 10ms^{-1} \\ 0.49 + 0.065WS_{10} & \text{for } WS_{10} > 10ms^{-1} \end{cases}$$

$$(2)$$

Figure 3. (a) Temperature cross section at  $77.6^{\circ}$  N on 01.01.2021. The black lines indicate the isotherms used to identify the warming events. The  $3^{\circ}$ C-isothermal is highlighted in green as an example and its easternmost point is marked with a round marker. (b) Time series of the on-shelf x-position of the easternmost points (the green marker for the example contour in (a)) for all isotherms during the winter season 2020/21. Those periods identified as a warming event are marked with color, corresponding to the value of the respective isotherm. The black arrow indicates the time of the cross section snapshot shown in (a).

**Figure 4.** (a) Current speed cross section at 77.6° N on 09.12.2019, with overlaid contour lines (black). Positions of peaks included in the analysis are indicated in color, peaks discarded due to insufficient total strength or prominence in light grey. The colored contours highlight the lowest contour line solely contributing to the peak marked with the corresponding color. The dotted vertical lines show the x-range associated with each current branch. (b) x-Positions of the geometric centers of each individual contour element contributing to the current branches shown in (a), plotted in the respective color. The locations of the discarded peaks are again shown in grey.

The total wind stress was decomposed in its zonal and meridional components ( $\tau_x$  and  $\tau_y$ , respectively), and subsequent Ekman transport was calculated from the individual components according to

$$M_x = \frac{\tau_y}{\rho_w f}; M_y = -\frac{\tau_x}{\rho_w f},\tag{3}$$

where  $\rho_w$  is the water density  $(1027 \mathrm{gkg}^{-1})$  and f is the Coriolis parameter, in turn depending on the Earth's angular velocity  $\Omega$  and the latitude  $\varphi$  according to  $f=2\Omega sin(\varphi)$ . In the following analysis, Ekman transport accumulated over 20 days (matching the threshold in the frequency filtering of the ocean data) is presented in order to account for time lags between wind forcing and Ekman transport as well as inertial impacts.

It shall be noted that in this paper, we use the term AW for any water mass originating in the WSC over the slope, that is warm and saline relative to the water masses present on the WSS. In particular, we do not define threshold values for temperature and salinity, as generally done in other studies.

Table 1. Overview of all warming events identified during the three winter seasons of the total study period.

| Warming<br>Event | Start    | End      | Duration<br>[days] | Total On-shelf<br>Heat Increase [10 <sup>15</sup> J] | Expanding<br>Coldest | Expanding Isotherms [°C] Coldest Warmest |  |
|------------------|----------|----------|--------------------|------------------------------------------------------|----------------------|------------------------------------------|--|
| 2019/20-A        | 9. Nov.  | 28. Nov. | 19                 | +559                                                 | 1.2                  | 4.2                                      |  |
| 2019/20-B        | 12. Dec. | 27. Dec. | 15                 | +807                                                 | -0.4                 | 2.8                                      |  |
| 2019/20-C        | 22. Dec. | 31. Jan. | 40                 | +3623                                                | -0.6                 | 4.4                                      |  |
| 2019/20-D        | 25. Feb. | 10. Mar. | 14                 | +261                                                 | 0.4                  | 3.4                                      |  |
| 2019/20-E        | 12. Mar. | 3. Apr.  | 22                 | +629                                                 | 0.4                  | 3.6                                      |  |
| 2019/20-F        | 27. Mar. | 25. Apr. | 29                 | +1737                                                | 0.0                  | 3.6                                      |  |
|                  |          |          |                    |                                                      |                      |                                          |  |
| 2020/21-A        | 22. Dec. | 14. Jan. | 23                 | +1479                                                | 0.4                  | 3.4                                      |  |
| 2020/21-B        | 2. Feb.  | 18. Feb. | 16                 | +129                                                 | -0.6                 | 2.4                                      |  |
| 2020/21-C        | 14. Feb. | 7. Mar.  | 21                 | +704                                                 | -1.2                 | 2.4                                      |  |
| 2020/21-D        | 8. Mar.  | 15. Apr. | 38                 | +971                                                 | -0.8                 | 2.4                                      |  |
|                  |          |          |                    |                                                      |                      |                                          |  |
| 2021/22-A        | 2. Dec.  | 21. Dec. | 19                 | +508                                                 | 0.2                  | 2.4                                      |  |
| 2021/22-B        | 28. Dec. | 21. Jan. | 24                 | +249                                                 | -0.8                 | 2.0                                      |  |
| 2021/22-C        | 9. Jan.  | 29. Jan. | 20                 | +541                                                 | -0.8                 | 2.0                                      |  |
| 2021/22-D        | 9. Feb.  | 26. Feb. | 17                 | +823                                                 | -1.2                 | 2.2                                      |  |
| 2021/22-E        | 22. Feb. | 17. Mar. | 23                 | +1460                                                | -0.6                 | 3.6                                      |  |
| 2021/22-F        | 15. Mar. | 1. Apr.  | 17                 | +1441                                                | -0.2                 | 3.8                                      |  |
| 2021/22-G        | 30. Mar. | 16. Apr. | 17                 | +418                                                 | 0.6                  | 4.0                                      |  |

## 3 Results and Discussion


Based on the final dataset prepared using the methods described in Section 2.2, we identify 17 individual warming events in the winter seasons of the study period. These include six events during the first winter (2019/2020), four during the second winter (2020/2021), and seven during the final winter (2021/2022), as summarized in Table 1. The events exhibit substantial variability in their duration, with the shortest lasting 15 days and the longest extending to 40 days from the start of the eastward expansion of the isotherms to the time they reach their maximum extent on the WSS. Also, the events differ in which isotherms are expanding, depending on the relative temperature difference between the intrusion and the ambient water on the WSS. The warming event magnitude, quantified by the total increase in relative heat on the shelf, ranges from +129 PJ to +3623 PJ, with an average daily heat increase of +41 TJ.day<sup>-1</sup> (standard deviation: 24.7TJ.day<sup>-1</sup>). In the following, we categorize these warming events based on their shelf-intrusion characteristics and examine the role of different cross-shelf exchange mechanisms in their development.

## 3.1 Ekman Transport







Persistent and strong southerly winds generate surface wind stress that drives an eastward Ekman transport. Goszczko et al. (2018) demonstrated how such conditions can facilitate the on-shelf transport of AW from the WSC at the surface, leading to a warming of the WSS. Examining the onset timing of all 17 warming events in relation to time series of surface wind stress, we identify four cases, 2019/20-A, 2020/21-A, 2020/21-C, and 2021/22-F, where this direct wind-driven forcing plays a substantial role. We present 2020/21-A as a case study. Figure 5 (a) illustrates the temporal and meridional development of the increase in heat content on the shelf during the warming event, revealing that the largest changes occur in the southern part of the study area. It can furthermore be seen that the strongest warming occurs approximately one week to ten days after the onset consistently across all latitudes.

Comparing this evolution to the atmospheric wind forcing in Figure 5 (b) shows that the warming event coincides with a pronounced southerly wind episode towards the end of December 2020. The resulting eastward Ekman transport accumulates to values of approximately  $20 \,\mathrm{m}^2\mathrm{s}^{-1}$ , the highest seen during the study period. The strong correspondence in timing underlines the role of Ekman transport as a key driver of cross-shelf exchange and subsequent shelf warming, with similar patterns evident in events 2019/20-A, 2020/21-C, and 2021/22-F (not shown).

Figure 5 (c) provides insight into the day-to-day progression of the AW intrusion onto the shelf, exemplified by the 3°Cisotherm at the 77.75° N-section. Comparison of the density of the intruding water at its initial entry point (locked into the circle on the easternmost extent in each isotherm) with the density profiles of the shelf water en-route shows how the shelf stratification determines the intrusion depth. While originally pushed towards the shelf at the surface due to wind-driven Ekman transport, the intruding AW encounters a much fresher and thus lighter shelf water mass, causing it to subduct and spread along the bottom as it moves eastward. The idealized comparison of the ambient shelf water density with only the initial intrusion density of course neglects the effects of mixing of the two water masses. However, the fact that such a simplified comparison fits the actual intrusion depth development so well suggests that the effects of mixing processes between intrusion and ambient shelf water are small over the course of the intrusion. In their work on the effect of a residual eddy overturning circulation, Tverberg and Nøst (2009) linked the depth of resulting AW intrusions to the tilt of the initial density front separating the Arctic-type shelf water from the AW carried northward by the WSC over the slope. The negative tilt of the black example contour (1027.6 - 1000 kgm<sup>-3</sup>) in Figure 5 (c), which deepens by approximately 90 m over a horizontal distance of 20 km, together with the intrusion along the bottom, align precisely with this relationship. This suggests that the connection can be generalized to intrusions driven by other cross-shelf exchange mechanisms, such as surface Ekman transport. In the end, Figure 5 (d) shows the total increase in heat content on the shelf at the same section due to the warming event. While the strongest warming eventually takes place along the bottom of the inner shelf areas, the path of the intruding AW can clearly be seen, originating from the surface at the shelf break, where it met the lighter shelf water and was forced to subduct.

The opposite process, when northerly winds drive offshore Ekman transport and induce upwelling at the shelf break, has been described by Cottier et al. (2007) based on data from the winter of 2005/2006. In that case, persistent northerly winds led to westward surface Ekman transport, which in turn triggered upwelling of AW from depth across the shelf break and

**Figure 5.** (a): Zonal- and depth-averaged heat content increase on the shelf as function of time and latitude for the warming event 2020/21-A. The dashed horizontal line indicates the latitude of the cross section shown in (c) and (d). (b): Time series of spatially averaged meridional wind stress (red axes and graphs) and resulting cumulative zonal Ekman transport (blue axes and graphs) during the same warming event period. (c): Day-to-day propagation of the 3.0°C-isotherm at 77.75° N over the course of the warming event (orange lines). The color-coding compares the initial potential density anomaly of the intrusion (locked in the circles on the isotherm contour lines) to the ambient water mass it is encountering along its way (in the background). The black line shows the 27.60 kgm<sup>-3</sup>-contour to illustrate the tilt of the density front. More detailed information about the composition of this panel is provided in Appendix C. (d): Zonal cross-section of the total heat increase on the shelf due to the warming event at the same latitude of 77.75° N.

onto the WSS. While this mechanism is also present in our dataset, its signature appears less pronounced compared to the onshore Ekman transport by southerly winds described above. While different warming events in all three winter seasons can be attributed to onshore Ekman transport, shelf warming due to upwelling induced by surface offshore Ekman transport is only seen in spring 2020. The most illustrative example is warming event 2019/20-F, shown in Figure 6. Even though other wind events with similar meridional negative wind stress forcing and magnitudes of offshore Ekman transport are present in the dataset, they do not lead to corresponding warming events. This suggests that upwelling-driven intrusions are less dominant and more susceptible to being masked by other concurrent processes. The interplay between different forcing mechanisms, including their potential synergistic or compensating effects on the warming of the WSS, is further examined in Section 3.3. Figure 6 (a) shows how warming event 2019/2020-F closely resembles the previous example 2020/21-A in terms of duration, magnitude, and temporal evolution. However, in contrast to 2020/21-A, the onset of 2019/20-F coincides with a particularly strong northerly wind event, further intensifying the already negative zonal (westward) accumulated surface Ekman transport (Figure 6 (b)). During this period, accumulated zonal Ekman transport reaches values of almost -40 m<sup>2</sup>s<sup>-1</sup>. Unlike the previous case, where the stratification forced AW to intrude at depth, the density of the water mass residing on the shelf prior to the warming event is higher than the density of the AW being lifted over the shelf break (Figure 6 (c)). This allows an intrusion further onto the shelf only at the surface, where in the end the strongest increase in heat content can be found (Figure 6 (d)).

In the context of discussing the impact of the shelf stratification, we present two more examples, highlighting further variations in the intrusion depths. Figure 7 (a) shows the day-to-day propagation of the 0.4°C-isotherm at 77.9° N during warming event 2021/22-D. The background color reveals that during the warming period, the shelf stratification is weak and zonally uniform (indicated by the two example isopycnals in Figure 7 (a)). Furthermore, there is no density front separating the shelf from the slope. This means that water masses from the shelf and the slope can readily exchange places in a so-called interleaving layer. This layer is not necessarily found at the surface or the bottom of the water column, but at the depth where the density of the intruding water exactly matches that of the ambient shelf water. In the presented example, this is the case at an intermediate depth of 100 m, where the intrusion propagates eastwards.

Furthermore, Figure 7 (b) illustrates how the intrusion depth may also change en route, if the intrusion encounters different and/or changing stratification regimes further east on the shelf. The initial horizontal density gradient between slope and shelf at the beginning of the presented warming event 2019/20-C is weak and the intrusion first penetrates the shelf at an intermediate depth range of 50 – 100 m (similar to the previous example 2021/22-D). However, the inner shelf area is occupied by a denser water mass. When the intrusion meets the corresponding front, it has to adjust its depth in order to match the ambient stratification and moves to the top of the water column isopycnally, from there on penetrating the shelf further east at the surface.

Taking all presented examples together, we conclude that the intrusion depth is governed solely by the relative density difference between the intruding water and the ambient shelf water, independent of the cross-shelf exchange mechanism responsible for triggering the intrusion in the first place. This underscores the crucial role of shelf stratification in determining the final intrusion depth.

Figure 6. Same as Figure 5, but for warming event 2019/20-F.

Figure 7. Same as Figure 5 (c), but for the warming events 2021/22-D and 2019/20-C

#### 3.2 WSC – STC – Interaction




In the previously discussed examples, cross-shelf water mass exchange and subsequent shelf warming has been examined primarily as a zonal process. However, it is clear from Figures 5 and 6 (a) that the resulting shelf warming events also exhibit a pronounced meridional component, with warming initiating in the south and propagating northward over time. This northward progression is particularly evident in warming event 2019/20-B, shown in Figure 8, where a distinct lag in warming onset can be observed along the shelf. While 2019/20-B provides the clearest example of this pattern, similar behavior is present, to varying degrees, in most of the 17 winter warming events identified throughout the study period. The detailed examination of the horizontal structure of warming event 2019/20-B in Figure 8 further reveals that the strongest increase in depth-averaged heat content on the shelf does not occur near the shelf break, but rather along the secondary shelf slope further onto the shelf, which separates Eggbukta from the shallower inner shelf regions (see Figure 2). This pattern of warming propagation strongly suggests that a topographically steered current serves as the primary mechanism for heat transport during this event. In the following, we investigate the connections between WSS warming events and the regional current systems in more detail.

Due to the diverging isobaths extending from the continental shelf break into Eggbukta (Figure 2 (b)), the WSC connects most strongly to the WSS in the southern part of the study area. As shown in Figure 9 (a), the WSC core composite maps of the three winter seasons illustrate how the WSC splits into two branches: a main northward-flowing branch along the steepest continental slope and a sub-branch that veers onto the WSS, following the secondary slope around Eggbukta as the STC.

To quantify the role of this topographic steering in the warming events, we analyze the relative strength of the STC compared to the WSC using the volume and heat transport ratios across the respective current branch sections (for our definition of the

**Figure 8.** Horizontal day-to-day development of the depth-averaged total heat content increase on the WSS during the ten first days of warming event 2019/20-B. The bathymetry contours have a 50 m-resolution.

current branch, see Section 2.2) at 77.6° N, shown in Figures 9 (b) and (c). Although differences in cross-sectional area and water mass characteristics complicate direct comparisons, the relative fluctuations in these transport ratios provide valuable insight. Many warming events coincide with increased STC volume and heat transport (2019/20-B, 2020/21-A, 2020/21-B, 2020/21-C, 2020/21-D, 2021/22-C, 2021/22-E, 2021/22-G), emphasizing the role of topographic steering in directing warm water from the continental shelf break onto the WSS.




The mechanisms controlling periodic STC intensification and associated warming events have been explored in previous studies. Nilsen et al. (2016) proposed that a strengthened WSC core leads to a widening of the current, enabling warm water masses from its eastern flank to reach shallower isobaths and be topographically guided onto the WSS. To investigate this connection in our data, Figure 9 (d) presents the WSC core speed further upstream at 77.35° N, revealing substantial variability with peak velocities ranging from 0.2 to 0.7 ms<sup>-1</sup>. Nilsen et al. (2016) linked increased WSC velocities and subsequent WSS warming to periods of strong southerly winds over the WSS and adjacent shelf break. These winds induce a positive meridional wind stress and a negative wind stress curl, and a subsequently converging eastward surface Ekman transport across the WSS (as discussed in Section 3.1). This leads to both a pile-up of water along the WSS break and the Spitsbergen coastline, with corresponding increases in the sea surface tilt, which in turn strengthens the northward currents underneath to maintain

geostrophic balance. However, a direct correlation between individual peaks in the WSC core speed and the cumulative zonal Ekman transport, shown in Figures 9 (d) and (e), is not immediately apparent. Instead, we observe that current speed variations tend to be less pronounced during periods of weak or negative zonal Ekman transport. This aligns with the general seasonal cycle, where WSC velocities are typically higher in winter due to the frequent passage of atmospheric storm systems, a pattern well documented in previous studies (e.g. Zahn et al., 2018; Wickström et al., 2020) but not analyzed in detail here.

Comparing the timing of the warming onsets with the time series of the WSC core speed shows that some warming events, such as event B in 2019/20, events A and B in 2020/21, as well as event B in 2021/22, coincide with WSC velocity peaks. However, other velocity peaks, such as those in January 2020 and November 2021, do not result in warming events, indicating that an increase in WSC speed alone does not necessarily trigger STC intensification and warming of the WSS.

Another factor influencing the intensification of the STC, which was not considered by Nilsen et al. (2016), is the variability in the exact position of the WSC. The spread of current core positions in the composites shown in Figure 9 (a) highlights a wide range of core locations, which translates into temporal variations in the WSC position at 77.35° N, as illustrated in Figure 9 (f). Parts of this variability can be attributed to higher-frequency processes such as eddy activity and baroclinic geostrophic adjustments to hydrographic changes, which were filtered from the data set using a 20-day-lowpass filter, see Section 2.2. However, the gray markers in Figure 9 (f) show that even the more stable current branch position (for details on the definition of branch and core position in this study, see also Section 2.2) exhibits substantial fluctuations. In particular, during nine periods within the three investigated winter seasons, the WSC branch position shifts closely toward or even crosses the shelf break. Eight of these occurrences coincide with increased volume and/or heat transport in the STC and the onset of warming events. This suggests that the lateral displacement of the WSC represents an additional mechanism to strengthen the STC, making it follow shallower isobaths and in that way drive the warming of the WSS. Similar to the reduced variability in the WSC peak speed, the position tends to shift toward deeper bathymetry during prolonged periods of negative zonal Ekman transport. In contrast, a connection between the WSC and shallower bathymetry feeding into the STC is more likely during episodes of weak or even southerly wind forcing. This also contradicts offshore surface Ekman transport due to northerly winds and subsequent upwelling of AW onto the WSS at depth as a shelf warming mechanism and could explain the corresponding weak connection discussed in Section 3.1.

# 3.3 Process Interaction and Additional Impact Parameters






The findings presented so far have focused primarily on the individual effects of different processes driving cross-shelf exchange and subsequent shelf warming. However, it has also become evident that multiple forcing mechanisms can interact and that a given warming event may result from a combination of these processes. For instance, as discussed in Section 3.2, the intensification of the STC and a subsequent warming event on the WSS can be triggered by either an increase in the WSC core speed, or an eastward displacement of the WSC core over shallower bathymetry, or by a combination of both. This interplay is particularly evident in winter 2020/21, where a certain co-variability between these two current properties can be observed (see Figure 9 (d) and (f)).

**Figure 9.** (a) Composites of core positions of the WSC over the continental slope and the STC on the outer WSS during the three winter seasons 2019/20, 2020/21 and 2021/22. WSC and STC core positions at 77.6° N are colored red and blue, respectively, and all core positions at 77.35° N are colored black. (b) – (f): Time series of the ratios between WSC and STC in the volume and heat transport, northward current core speed, cumulative zonal Ekman transport (spatially averaged over the whole WSS break area, see Figure 2 (b)) and WSC position throughout the three winter seasons. Note the logarithmic y-axis scales in (b) and (c). At 77.35° N, the shelf break (x=0 km, indicated with the dashed horizontal line in (f)) is located at 340 m depth. The onset times of all warming events are indicated with black arrows on top of (b). The gaps in (c) result from negative heat transport values, as the temperatures of the water mass transported by the STC on the shelf drops below the reference temperature of 0°C. In (f), WSC core positions are marked with black markers, while branch positions are marked grey (see Section 2.2 for details on the definitions). Core positions during times when the core depth is found below 150 m are colored in purple.

Similarly, atmospheric wind forcing can contribute to WSS warming events through both direct and indirect ways. Directly, strong southerly winds induce on-shelf Ekman transport, while indirectly, they modify the sea surface tilt, which can subsequently accelerate the WSC. To illustrate how different processes can interact, Figure 10 presents the warming event 2020/21-C as an example of a shelf warming event likely driven by multiple mechanisms. Comparison of Figure 10 (a) and (b) reveals that the warming starts at a time when the WSC core is located on the shelf (positive x), indicating a direct input of warm AW into the STC. Even though the core position moves westward during the following days, the branch as a whole shifts towards shallower bathymetry. At the same time, the WSC core speeds up to almost 0.6 ms<sup>-1</sup>, with the peak speed coinciding with the largest daily increase in heat content on the shelf across all latitudes (Figure 10 c)). Furthermore, while wind forcing is relatively weak at the beginning of the warming event, a strong wind event occurs at the same time (Figure 10 (d)). Although this wind event cannot have caused the initial WSC speed-up, the associated Ekman transport, reaching up to 10 m<sup>2</sup>s<sup>-1</sup>, likely contributes to the overall warming by facilitating additional cross-shelf advection of warm water being already present near the shelf break. This example underscores how different forcing mechanisms may not only act independently, but also amplify one another, emphasizing the complexity of cross-shelf exchange and shelf heat variability in this region.

Beyond the individual forcing mechanisms discussed so far, the warming event 2020/21-C highlights yet another important factor influencing the WSS warming: the variability in the temperature of the AW transported northward by the WSC. As the primary heat source for all cross-shelf warming events, fluctuations in the temperature of the WSC can substantially impact the magnitude of warming events on the WSS. Although this variability has been extensively studied in the context of northward heat transport toward the central Arctic Ocean (e.g. Schauer et al., 2004; Muilwijk et al., 2018) and the warming of western Spitsbergen fjords (e.g. Pavlov et al., 2013; Tverberg et al., 2019), its role in driving the intensity of the WSS warming has, to the authors' knowledge, not been explored before.

Figure 10 (e) shows the average conservative temperature of the upper 300 m of the water column over the slope (x<0 km) during the warming event. A notable increase of more than 0.5°C is observed, coinciding with the increase in heat content on the shelf. The northward propagation speed of this temperature anomaly (0.2 – 0.4 ms<sup>-1</sup>) approximately matches the WSC speed. A broader analysis of this relationship is provided in Figure 11, which presents the time series of the upper 300 m average conservative temperature, absolute salinity and potential density anomaly over the slope at 77.6° N (solid, colored curves) throughout the study period. Figure 11 (a) reveals that several warming events (2019/20-C,-D,-F, 2020/21-A,-C, 2021/22-D,-E,-G) coincide with rising temperatures or peaks in the temperature time series, further supporting the role of upstream hydrographic variability in the WSC in modulating WSS warming events.

A particularly striking example is the warming event 2019/20-C, which stands out as the longest and by far strongest warming event identified in this study period (see Table 1). From Figure 11 (a) and (b), it becomes evident that this event closely coincides with a pronounced temperature and salinity increase in the WSC region over the slope. Unlike other warming events, there is no substantial increase in the STC/WSC heat and volume transport ratios, no eastward displacement of the WSC (Figure 9), and no substantial zonal Ekman transport anomalies observed. Instead, the increase in temperature over the slope leads to a density decrease (the salinity increase does not compensate for the temperature increase) and a subsequent degeneration of the density front separating the shelf from the slope. This allows warm AW from the WSC to penetrate the

Figure 10. (a) Zonal- and depth-averaged heat content increase on the shelf as function of time and latitude during the warming event 2020/21-C. (b) Time series of the WSC core (black markers) and average branch position (light gray markers) at  $77.35^{\circ}$  N. The shelf break at x = 0 km is highlighted with a black dashed line. (c) Time series of WSC northward core speed at  $77.35^{\circ}$  N. (d) Time series of spatially averaged meridional wind stress (red axes and graphs) and resulting cumulative zonal Ekman transport (blue axes and graphs). (e) Average temperature of the upper 150 m of the water column over the slope (x

**Figure 11.** Average conservative temperature (a), absolute salinity (b) and potential density anomaly (c) of the upper 300 m of the water column over the slope (x<0 km, colored solid lines) and the shelf (x>0 km, black dashed lines) at 77.6° N. In panel (c), the gray shading furthermore indicates the range of the average surface and bottom potential density anomaly on the shelf. The onset times of all warming events are indicated with black arrows on top of panel (a).

WSS in an interleaving layer at intermediate depth (see Figure 7 (b)). As a consequence to the reduced zonal density gradient between the shelf and slope, also the baroclinic component of the WSC weakens. As the slope region warms further and ultimately becomes lighter than the shelf (Figure 11 (c)), also the baroclinic current structure reverses, resulting in a downward intensification of the northward flow in the WSC. As a consequence, the core does not only accelerate (Figure 9 (d)), but also shifts to a depth of approx. 400 m in the water column during this period (Figure 9 (f)). Thus, in this case, the increased core speed of the WSC during the warming is not a driver as described in Section 3.2, but rather a consequence of the altered zonal density gradient.

The impact of the warming event 2019/20-C on the WSS extends well beyond its immediate duration. As seen in Figure 11 (a) and (b), the shelf does not only experience a temperature increase of approximately 2°C, but also becomes flooded with saline water (salinity up to 35 g/kg) that persists throughout the remainder of the winter season and leads to an increase in shelf water density. In fact, over the course of a winter, the shelf becomes progressively denser not only due to saline AW intrusions, but also due to surface heat loss and cooling of the shelf water column, and, if freezing temperatures are reached, by an increase in salinity due to brine release from local or upstream sea ice formation. This seasonal evolution of the shelf hydrography plays a crucial role in determining the depth of warm water intrusions. Early in the season, when the shelf is still relatively warm and fresh, warm intrusions are more likely to occur at depth, whereas later in the season, when the shelf has cooled and become more saline, warm AW intrusions reach the surface more frequently. In the case of the warming event 2019/20-C, the corresponding AW intrusion resulted in a substantially warmer and more saline shelf during the second half of winter 2019/2020. This marks a regime shift, from all warming events occurring along the bottom earlier in the season, while those later in the season penetrate the shelf at the surface rather than at depth. These findings underscore how cross-shelf intrusions shape the seasonal and inter-annual variability of the shelf hydrography.

## 4 Summary, Conclusions and Outlook







The present study investigates the causes and dynamics of 17 warm AW intrusions onto the WSS in winter 2019/20, 2020/21, and 2021/22, as identified from high-resolution regional model data. A series of previous studies have addressed individual forcing mechanisms based on observational data sets and idealized model simulations. This study builds on these findings by examining, for the first time to our knowledge, the interplay of multiple processes driving cross-shelf exchange within a fully dynamical regional ocean model. This integrated approach enables not only a direct comparison of the effectiveness of different mechanisms, but also highlights both synergistic and counteractive effects that emerge from their interaction.

Atmospheric forcing plays a crucial role in triggering and modulating warming events. Strong and persistent southerly winds over the continental shelf break can drive on-shelf Ekman transport, directly pushing AW onto the shelf. There is also some evidence of an opposite mechanism, where northerly winds drive offshore Ekman transport, leading to upwelling of AW from deeper parts of the continental slope and across the shelf break. However, this upwelling-induced warming appears to be less pronounced than the direct on-shelf Ekman transport mechanism, presumably due to the fact that the WSC position tends to be shifted over deeper bathymetry during these periods of persistent northerly wind, hence warm AW is less likely to reach the shelf.

Many warming events exhibit a meridional progression, beginning in the south and propagating northward. This pattern is particularly evident for warmings of the WSS associated with an intensification of the STC, a branch of the WSC that follows shallower isobaths around Eggbukta and continues into the Isfjorden Trough. STC intensifications can be triggered either by a shelf-ward displacement of the WSC core or by an increase in WSC core speed. While these two current properties are interlinked, no direct connection between sea surface tilt variations caused by direct on-shelf Ekman transport and WSC speed variations is evident. However, the amplitudes of WSC velocity peaks appear to be reduced during prolonged periods of strong

**Table 2.** Summary of all warming events identified during the three winter seasons of the total study period and related forcing mechanisms.

| Warming   | Total On-shelf                     | Direct             | Ekman     | STC Intensific | ation due to WSC  | WSC Temperature | Final              |
|-----------|------------------------------------|--------------------|-----------|----------------|-------------------|-----------------|--------------------|
| Event     | Heat Increase [10 <sup>15</sup> J] | Ekman<br>transport | Upwelling | Speed Increase | Core Displacement | Increase        | Intrusion<br>Depth |
| 2019/20-A | 559                                | X                  |           |                |                   |                 | Bottom             |
| 2019/20-B | 807                                |                    |           | x              | X                 |                 | Bottom             |
| 2019/20-C | 3623                               |                    |           |                |                   | X               | Surface            |
| 2019/20-D | 261                                |                    | X         |                |                   | X               | Surface            |
| 2019/20-E | 629                                |                    | X         | X              |                   |                 | Surface            |
| 2019/20-F | 1737                               |                    | X         |                |                   | X               | Surface            |
| 2020/21-A | 1479                               | x                  |           | X              | X                 | X               | Bottom             |
| 2020/21-B | 129                                |                    |           | X              | X                 |                 | Bottom             |
| 2020/21-C | 704                                | X                  |           | X              | X                 | X               | Bottom             |
| 2020/21-D | 971                                |                    |           |                |                   |                 | Bottom             |
| 2021/22-A | 508                                |                    |           |                | X                 |                 | Bottom             |
| 2021/22-B | 249                                |                    |           |                |                   |                 | Bottom             |
| 2021/22-C | 541                                |                    |           |                | X                 |                 | Bottom             |
| 2021/22-D | 823                                |                    |           |                |                   | X               | Intermediate       |
| 2021/22-E | 1460                               |                    |           |                | X                 | X               | Surface            |
| 2021/22-F | 1441                               | X                  |           |                |                   |                 | Surface            |
| 2021/22-G | 418                                |                    |           |                | X                 | X               | Surface            |

offshore Ekman transport, and the current tends to be positioned further offshore over deeper bathymetry. This suggests that in recent winters, with more frequent southerly wind events in eastern Fram Strait, the WSC has been more likely to connect with shallower bathymetry and feed into the STC.

Individual warming events are often influenced by multiple cross-shelf exchange mechanisms acting simultaneously. An overview of the contributions of different processes to each warming event is provided in Table 2. Based on the 17 warming events analyzed in this study, it is not yet possible to determine whether any single mechanism consistently leads to stronger warming events than others, particularly given that many events result from combined forcing. However, it can be seen that WSC core displacement is the most frequently involved mechanism triggering WSS warming events, and furthermore the strongest warming events tend to coincide with periods of increased temperature in the AW source waters transported northward by the WSC, highlighting the importance of upstream oceanic variability in modulating shelf warming magnitudes.

Table 2 also highlights that some warming events, including the strongest observed event (2019/20-C), cannot be directly linked to any of the cross-shelf exchange mechanisms discussed in this study. An alternative explanation for such events is the residual eddy overturning, as proposed by Tverberg and Nøst (2009). Our analysis is based on a frequency-filtered dataset (lowpass-filtered with cutoff frequency 20 days) designed to isolate net warming effects on multi-week timescales, therefore not suited to directly assess eddy activity. Such an investigation based on the unfiltered model output would furthermore be beyond the scope of this study. However, our filtered data set captures the residual effects of the eddies, in other words, the overturning described by Tverberg and Nøst (2009). Given the frequent occurrence of barotropic and baroclinic instabilities along the eastern flank of the WSC (Teigen et al., 2010, 2011), it is likely that eddies play a significant role in the triggering of warming events on the WSS. This is further supported by the consistency between our findings on stratification-controlled intrusion depth and the results of Tverberg and Nøst (2009). Future studies using the same original ROMS simulations, but focusing e.g. on a period around the warming event 2019/20-C and explicitly considering eddy dynamics, could provide further insights into their role in shaping heat transport in this region.

Another key finding is that the depth at which AW intrudes onto the shelf is determined primarily by the local shelf stratification. AW will penetrate isopycnally at the density level that matches the ambient shelf water masses, regardless of the specific cross-shelf exchange mechanism that initiated the intrusion. If horizontal density gradients exist on the shelf, the intrusion depth will follow the isopycnals and may dynamically adjust as the intrusion progresses. Consequently, the seasonal evolution of shelf hydrography strongly influences the nature of warming events. In early winter, when the shelf is still relatively warm and fresh, intrusions tend to occur at depth (see Table 2). As the season progresses, cooling and an increase in salinity, driven by surface heat loss, prior AW intrusions, potential brine rejection from local sea ice formation, or advection of brine-enriched water transported by the SPC make surface intrusions more frequent.

The comprehensive analysis of different cross-shelf exchange mechanisms and subsequent WSS warming events presented in this study is only possible based on model data with sufficient resolution in space and time. However, from validation against observations (see Appendix B), it can be seen that the model data exhibit biases that must be considered when interpreting the results. Across the WSS break, model temperatures are found to be too low, especially on the shelf in early winter. This discrepancy could have implications for the representation of early winter warming events, potentially leading to an overestimation of their strength in the simulations. Furthermore, the simulated salinity of the AW in the core of the WSC is consistently too low compared to observations. In view of the recent freshening of the AW since 2018 (described e.g. by Kolås et al. (2024)), this is an interesting finding, as the simulations might be more representative for future than current conditions. As biases in temperature and salinity largely compensate each other when it comes to the resulting density fields, the model demonstrates a very good representation of topographically steered currents, both in terms of speed and direction, indicating that key dynamical features governing cross-shelf exchange are realistically simulated.

From a broader regional ocean climate perspective, the cross-shelf exchange bringing AW onto the WSS represents only the initial step in its potential pathway toward the fjords along the West Spitsbergen coastline. For AW to reach these fjords, especially Isfjorden, which lacks a separating sill and is therefore particularly susceptible to warm water inflow, it must first spread across the shelf, overcoming additional barriers. One such barrier is the SPC, which effectively shields the fjords from

direct AW influence. Previous studies have identified wind forcing as a key mechanism controlling the opening of the fjords to warm water intrusions (Nilsen et al., 2016; Fraser et al., 2018; Skogseth et al., 2020; De Rovere et al., 2024). Furthermore, the density contrast between the intruding AW and the fjord interior modulates the inflow (Nilsen et al., 2008; Tverberg et al., 2019; Skogseth et al., 2020), in a manner similar to the frontal stratification control of the intrusion depth described in this study for the WSS break.



Future research will focus on these processes impacting the subsequent distribution of AW on the WSS and towards Isfjorden. Using Lagrangian particle tracking simulations based on the unfiltered ROMS current fields produced for this study, the pathways and conditions that ultimately allow AW to enter the fjord will be investigated to gain a more detailed understanding of the regional heat transport into Isfjorden and in that way an increased knowledge of the air-ocean heat flux variability in the fjord system.

# **Appendix A: Frequency Filtering**

# Appendix B: Model Validation





The ROMS setup used as a basis for this study has been newly implemented and has not been used in this configuration before. Therefore, we provide a brief validation against available observations to assess its performance. The validation is based on measured data from the ocean observation program at UNIS, which covers Isfjorden and the adjacent shelf areas (Skogseth et al., 2020). Spatial coverage is ensured by regular research cruises, which provide hydrographic and current measurements along dedicated sections. Since the WSS region and in particular the WSC over the continental slope are only covered once per year during autumn in the last decade, observations directly within our study area and period are limited to a single annual cross-section (Skogseth, 2025a, b, c). To complement this, we additionally utilize data from a mooring deployed at the mouth of Isfjorden (78.0607° N, 13.523° E) (Skogseth and Ellingsen, 2022; Skogseth, 2024), to also assess the temporal variability of the model on time scales similar to those investigated in the main study. The locations of the CTD/lowered-ADCP stations used in the validation, as well as the mooring site, are shown in Figure B1.

#### **B1** Shelf Break Section

To validate the model simulations within the main study area, we use hydrographic (CTD) and current (lowered ADCP) data from three research cruises along a zonal cross-section across the western Spitsbergen shelf (WSS) break at 78.0° N (Skogseth, 2025a, b, c). For a direct comparison, the frequency-filtered ROMS simulation data from the time steps matching the observations are co-located both horizontally and vertically to the measurement locations using linear interpolation. The resulting cross-sections of conservative temperature, absolute salinity, in-site density and northward current speed are shown in comparison to the respective observations in Figure B2.

For the conservative temperature, the results indicate a systematic bias in the simulations, where surface waters tend to be too cold, while deep waters over the slope appear too warm. This tendency becomes more pronounced over the course of the simulation period, suggesting that the model exhibits enhanced vertical mixing, which erodes vertical gradients — a known issue in ROMS. On the shelf, the modeled temperatures match the observations remarkably well in autumn 2019 but are too cold in 2020 and 2021.

The simulated salinity fields also reveal biases, with a surface layer that is consistently too fresh, particularly over the shelf in 2020 and 2021. Furthermore, the AW over the continental slope is systematically too fresh in all three years. Despite these biases in temperature and salinity, the resulting density fields show excellent agreement between the model and observations, suggesting that the errors in temperature and salinity largely compensate for each other. This is an encouraging result for the simulation of ocean dynamics. Indeed, the observed current cross-sections from the UNIS cruises in all three years are well captured by the model, strongly supporting its ability to represent the regional circulation patterns.

Figure A1. Time series of (a) temperature, (b) salinity and (c) meridional current speed at an example point from the WSS break section at  $77.9^{\circ}$  N at x=5 km in 10 m depth. The light curves in the background display the raw daily values, the opaque curves on top show the low-pass filtered smooth time series used for the further analyses.

**Figure B1.** Positions of the  $78.0^{\circ}$  N-section across the WSS break as used in this study (black line), together with the positions of relevant CTD/lowered-ADCP stations (blue points) covered as part of the operational UNIS ocean observation program. The location of the mooring in the mouth of Isfjorden is shown with a red star.

**Figure B2.** ROMS-simulated hydrography at the  $78.0^{\circ}$  N-section compared to observations (hydrography: CTD profiles, currents: lowered ADCP) in autumn before each of the three winters seasons investigated in this study.

# **B2** Mooring







To complement the cross-section comparison based on annual CTD profiles (see previous Section B1), we additionally validate the model simulations using time series from a mooring deployed at the southern side of the mouth of Isfjorden (Skogseth and Ellingsen, 2022; Skogseth, 2024). This mooring, located at 78.0607° N, 13.523° E, provides the closest continuous observations to our study area over the WSS break, offering valuable insight into temporal variability on sub-annual time scales. Observational data are available at hourly resolution for two periods: autumn to autumn in 2020/21 and 2021/22. For direct comparison, the unfiltered hourly model data from the mooring location were vertically interpolated to match the measurement depths. Furthermore, both observed and simulated time series were detided and frequency-filtered using the same processing methods applied to the data set in the main study.

A comparison of simulated and observed conservative temperature, absolute salinity, in-situ density, and absolute current speed is shown in Figure B3. The model generally reproduces the temperature variations well, with very good agreement during summer. However, simulated temperatures tend to be slightly lower than observed, particularly in early winter, and exhibit reduced variability in the unfiltered hourly data. The overall temperature distribution, including its range and peak locations, closely matches the observations. Salinity comparisons reveal no significant systematic biases, except for a slightly lower salinity in winter, which corresponds to the negative temperature bias described above. As with temperature, salinity variability appears to be somewhat dampened in the model.

Despite these minor discrepancies in temperature and salinity, the density time series show good agreement between model and observations. The influence of the less-saline winter water masses in the model is visible, but otherwise, the modeled and observed density curves align very well, with increased variability reflecting the corresponding temperature and salinity fluctuations. The overall distributions are very similar. A similar pattern is seen in the current speed comparison, where variability between the model and measurements is well captured, with the main difference being a slight overestimation of high current speeds in the model.

The overestimation of high current speeds observed in the time series in Figure B3 (g) is also evident in the current roses shown in Figure B4. However, the general inflow pattern into Isfjorden at the mooring location is well represented in the simulations, indicating that the model captures the dominant circulation features well. The strong influence of bathymetry on current directions throughout the water column is clearly reflected in both observations and simulations. With the mooring positioned at a bottom depth of 200 m and measurements taken at 65 m depth (hydrography in 2020/21 at 54 m), the model accurately resolves topographic steering effects, ensuring a realistic representation of the observed flow. Given this strong agreement, similar model performance can be expected further out on the WSS and along the shelf break, supporting the reliability of the simulated current dynamics in the broader study region.

Figure B3. Comparison of observed and simulated conservative temperature (a), absolute salinity (c), in-situ density (e) and absolute current speed (g) at the location of the mooring in the mouth of Isfjorden. The light curves in the background display unfiltered hourly values, the opaque curves on top show the low-pass filtered smooth time series similar to the data used in the main study. The overall data distributions of the respective variables are shown in the right side panels (b) - (h), with colors matching the graphs in (a) - (g).

**Figure B4.** Comparison of current roses between mooring observations and ROMS simulations, based on the full two-year time period. The centers of the current roses are placed at the actual mooring location. The bathymetry is shown as gray contour lines in the background (50-m-spacing, fading with depth).

#### **Appendix C: Density Control of the Intrusion Depth**




In Figures 5 (c), 6 (c) and 7 of the main study, we illustrate the spatio-temporal evolution of a warm-water intrusion onto the WSS. The figures especially highlight the depth adjustment of different intrusions according to the relative density difference between the intruding water masses and the ambient water masses on the shelf. The plots are composites of data of different origin, and here we explain their composition in more detail with the help of Figure C1.

In Step 1, the initial position of the example isotherm representative for the warming event is shown. The potential density anomaly at the depth and x-position of the easternmost point at the respective time step gets locked in the little circle. In Step 2, the day-to-day propagation of the isotherm is added, while the initial potential density anomaly extracted in Step 1 stays locked in the small circles. In Step 3, the potential density profiles at the easternmost point of each isotherm are extracted from the overall dataset for the respective time step. In cases when the isotherm progresses further than 1 km over one day, a gap is seen between the profiles. In Step 4, these gaps are linearly interpolated. Given that the profiles are not from the same, but from consecutive time steps, this interpolation is in fact an interpolation not only in space, but also in time. It shall be noted that along the bottom, the interpolation lacks data from shallower profiles further up on the shelf, hence smaller data gaps remain. In Step 5, the background data west of the circle on the initial isotherm is filled with data from the initial time step. Finally, the composite is completed in Step 6 by filling the background data east of the circle on the final isotherm with data from the final time step. The example potential density contour shown in black is based on the final composite.

**Figure C1.** Schematics illustrating the composition of Figures 5 (c), 6 (c) and 7 in the main study.

Data availability. The re-gridded and frequency-filtered ROMS model data set together with the atmospheric wind forcing time series used for the analyses presented in this study have been published at https://doi.org/10.5281/zenodo.15188605 (Frank and Albretsen, 2025). The full simulations are stored at the IMR data servers and can be made available upon request.

Author contributions. LF is responsible for the the main analysis and the preparation of this manuscript. The ROMS model simulations were set up, run and validated by LF and JA. RS prepared the observational data sets for the validation. All authors contributed with discussion of the results and reviewing of the manuscript.

Competing interests. The authors declare that they have no conflict of interest.

Acknowledgements. The authors acknowledge the Norwegian Research Infrastructure Services (NRIS) for providing the high-performance computing resources used for the ROMS simulations. We thank MET Norway, in particular Marta Trodahl, for providing the model grid, and Louise Steffensen Schmidt at the University of Oslo for making the runoff dataset available. Additionally, we acknowledge the use of ChatGPT for assistance in figure production and the formulation and language improvements in individual paragraphs in this paper.

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
