# Peer review of "Mechanisms of Warm-Water Intrusions onto the West Spitsbergen Shelf during Winter"

_EGUsphere, 2025_

## Author Comment (AC1)

**General comment**

This paper utilizes outputs from an ocean model to investigate mechanisms of Atlantic Water (AW) intrusions onto the West Spitsbergen Shelf (WSS) during winter. Authors describe different mechanisms, principally focusing on the influence of winds, characteristics of the Atlantic current, interactions among different water masses, and the interplay between these factors. I found the topic of this paper relevant and innovative, considering the current state of literature. To my knowledge, this is the first instance of an ocean model being used to assess the mechanisms behind different warming events on the WSS. The methodology adopted in this study is sound and appropriate to address the scientific questions. Even though this paper does not give a strong final message (this is understandable given the complexity of the system under study and the objective of the paper), I believe this work is informative and worth of publication.

However, I have concerns regarding the clarity and structure of the manuscript, which could be improved. I recommend the authors enhance the overall clarity of the manuscript by using a clear and concise language to effectively communicate the methodology, results and main findings. Consider integrating key information regarding the methodology into the main text rather than regarding all details to the appendices. I suggest the authors to more explicitly highlight the novel contributions of this study in relation to existing literature, particularly in the Summary and Conclusions section. A clear articulation of what is new and how it builds upon previous knowledge will help understand the significance of the findings.

Please refer to my minor comments for specific issues.

$\Rightarrow$ Thank you for this very positive and encouraging feedback. Especially the fact that there is a large agreement between the comments from you and the other referee review, provides a good basis for the revision of the manuscript. We have highlighted the novelty of our study as suggested and moved large parts of the information on the data processing from the Appendix to Section 2.2 (Data Processing and Analysis Methods) in the main body of the manuscript. We hope that we could satisfactorily address all points raised.

**Minor comments**

Line 4: did you mean "excess" instead of "access"?

$\Rightarrow$ Yes, thanks for spotting this mistake.

Line 5: substitute "full" with "fully"

⟹ Done.

Line 12: consider substituting "near-surface" to "surface-layer"

⟹ Yes, thank you for this suggestion.

Lines 15-16: Including future research perspectives in the Abstract is out of scope, particularly since authors discuss fjords, but the paper is focused on the WSS and shelf break. I suggest the authors to keep this only in the final section of the manuscript.

⟹ We have removed the outlook from the Abstract and keep it to the respective Section 4 of the main manuscript now.

Line 27: Please add a reference for the statement "The NwAFC and the WSC merge west of Svalbard around 78∘ N"

⟹ Done.

Lines 30-31: Consider rephrasing the sentence to improve readability: move "typically" at the beginning of the sentence, substitute "of which one" with "one of which". It is worth noting that this refers to the western branch of the WSC, as the eastern branch is introduced later.

⟹ Thank you for this detailed and very helpful comment, we have revised the manuscript accordingly.

Lines 40 and 41: Please provide references for the SPC and ECS.

⟹ Done.

Line 48: Consider replacing "restricts" with "prevents"

⟹ Done.

Lines 55-57: improve the clarity and readability of this sentence

⟹ We have re-written this part and hope it is more understandable now.

Figure 1: The figure appears well-designed, but the names of the Atlantic-type currents are hard to read against the blue background. Using bold font for these labels may enhance visibility. Additionally, please consider adding ticks for longitude and latitude values. The label '1' is not clearly connected to the coastal current; consider adding the 'SPC' label on the map where there is available space, along with a black line connecting it to the coastal current.

⇒ We have added small boxes around the labels to increase the readability. We have also added latitude and longitude ticks, and link labels for SPC and STC directly using lines rather than using the number.

Line 65: add "the" before "slope"

⇒ Well spotted, thank you.

Lines 77-79: Please consider improving readability of this sentence, as it is key to the present paper. Use the common forms as "gap of knowledge" and "This study aims …".

⇒ We have adjusted this sentence using the phrasing you suggested. Thank you for the suggestions.

Lines 81-82: This sentence regarding Isfjorden seems out of context.

⇒ We have moved this part into Section 2, where it indeed fits much better together with other information on how the data set was limited in space and time for the study.

Lines 81-84: I suggest to quickly recall the motivations why the authors focus the interest on winter months: sea-ice, water column stratification, preconditioning to summer, etc.

⇒ See reply above, this has been moved to Section 2 as well, and a reminder about the motivation has been added according to your suggestion.

Line 98: delete "for our purposes"

⇒ Done.

Figure 2: Add info regarding depth contour levels.

⇒ Done.

Lines 104-105: What does it mean "limited to"? Did you mean that the simulation ran from April 2019 to October 2024 but you considered only winter periods in the present analysis?

⇒ Yes, and we have re-written this sentence to make this clearer.

Line 110: Please consider adding a few lines at the end of this paragraph to summarize the key aspects of the model validation: key strengths and weaknesses, etc..

⇒ Done.

Lines 122-126: These sentences are difficult to understand, please improve their clarity.

⇒ We have rewritten the sentences to improve the readability.

Line 142: Consider adding here some key details characterizing this detection method. What are the temperature/salinity and eastward movement thresholds (i.e., minimum temperature/salinity anomaly and distance) that defined a warming event? Clarify when a warming event ends.

⟹ As suggested by your overall comment, we have moved the details on the warming event identification to Section 2.2 in the main part of the manuscript and revised it accordingly.

Line 165: delete "actually"

⟹ Done.

Lines 167-168: as these heat content values are relative to a previous state, I recommend adding a + in front of these numbers (also throughout the rest of the text and figures)

⟹ Yes, good idea, we have changed this throughout the manuscript.

Line 201-202: Can you add some numbers supporting this statement? For example, comparing the range of maximum heat content increases reached in different warming events for upwelling events vs onshore Ekman transport events.

⟹ This statement refers rather to the number of occurrences, and we have rephrased and added some numbers to clarify this.

Line 203: Add "meridional" to "negative wind stress"

⟹ Done.

Lines 209 and 210: add "zonal" after "negative" and after "accumulated"

⟹ Done.

Figure 3 (and similar figure 4):

- Consider moving the density colorbar to the left, adjacent to panel c, for improved reference.

- Caption (c): add "(orange lines)" after "…one specific isotherm…". I have difficulties understanding how density between different orange lines was calculated. Please consider simplifying this part of the caption.

- Caption (d): what is the final heat increase? Does this refer to the heat increase at the last time step of the warming event?

- I see there are no data in 3c and 4c density sections in some near-bottom locations, whereas heat content increase data (3d and 4d) are present, why? Clarify this discrepancy.

- The interpretability of figures 3c, 4c and 5 is difficult and not immediate. Here is a suggestion for the authors: plot 2 panels, one for the initial and another for the final time step of the warming event, each showing sections of density anomalies compared to the WSC core. This may lose the temporal information about the eastward progression of the AW along the shelf, but it may point directly to the importance of the density difference between AW and shelf ambient waters.

⇒ Thank you for these very detailed and helpful comments regarding this Figure. We were aware from the beginning that especially panel (c) can be difficult to explain. We have improved the structure of the figure by moving the color bars as you have suggested. We have streamlined the figure caption, while at the same time adding extensive explanation about the composition of panel (c) in the Appendix. We would really like to stick to this presentation, and hope the additional explanation in the Appendix allow for this.

Figure 6 caption: add definition of depth contour levels

⇒ Done.

Line 262: Is "decreasing" correct? I believe authors meant "increasing".

⇒ We believe "decreasing" is correct: Southerly winds are typically stronger further west, farther away from land in central Fram Straight. Therefore, eastward Ekman transport is also larger in the west and decreases towards the shelf and the Spitsbergen coastline. In any case, we have slightly reformulated this sentence to avoid confusion.

Lines 280-283: Is it possible to quantify the occurrence and significance of such events? The current text is vague, using terms like "during certain periods", "frequently", "often". I suggest being clearer about the real influence of this mechanism, if possible, as the authors argue that this represents an additional mechanism for WSS warming.

⇒ Yes, thank you for this valuable input. We have strengthened our point by specifically providing numbers to the occurrences.

Lines 284-287: I suggest rewriting this sentence to render it clearer, as currently it is too long and complex.

⇒ We have split this sentence into smaller parts and slightly rewritten it to increase the readability.

Lines 293-296: I suggest recalling a figure for the reader to consult for further clarification.

⇒ We have added references to both Figure 10 and the respective section of the manuscript.

Figure 7: Please adjust the y-axis labels of panels d, e and f to prevent overlap. Add description of grey line in panel f.

⇒ Done.

Figure 8 caption: Clarify the significance of black dashed line in panel b

⇒ Done.

Figure 9 caption: Please add panel names following the parameters listed.

⇒ Done.

Table 2: this table would benefit from including the total on-shelf heat increase from Table 1. This addition would provide readers with a quick quantitative comparison of the different events and mechanisms. It would also say something more about the magnitude of those events lacking a clear driving mechanism.

⇒ Yes, very good point, adding the column on the total heat content increase helps to compare the effects of the different forcing processes. Thank you for this suggestion, we have implemented it in the revised manuscript.

---

## Author Comment (AC2)

**General Comments**

This manuscript presents a well-executed and valuable contribution to our understanding of Atlantic Water intrusions onto the West Spitsbergen Shelf during winter. The authors are commended for their thorough analysis, detailed discussion of mechanisms, and extensive reference list. The modeling work is clearly presented and captures a range of relevant cross-shelf exchange processes with appropriate nuance and attention to seasonal variability. That said, several key methodological details currently placed in the appendices—particularly those relevant to the identification of the Atlantic Water core and diagnostics of shelf-slope dynamics—should be moved to the main methods section to ensure clarity and reproducibility. Some discussion of mixing processes and their role in modulating heat transport and density structure would further strengthen the work. Overall, I find the manuscript well-prepared and recommend publication after technical corrections.

⇒ Thank you for this positive and very encouraging feedback. Especially the fact that there is a strong agreement between the comments from you and the other referee review, provides a good basis for the revision of the manuscript. Besides moving information on the data processing from the Appendix to Section 2.2 (Data Processing and Analysis Methods) in the main manuscript, we have added some discussion on the effect of mixing for the intrusions. We hope that we could satisfactorily address all points raised.

**Specific Comments**

**Title:** Consider adding "Mechanisms of…" to the beginning

⇒ Thank you for this suggestion, we believe it adds valuable information to the title and makes it more interesting. We have adjusted our manuscript accordingly.

**Line 4:** What is "access heat"? Excess? Please clarify or revise.

⇒ Yes, it should be "excess". Thank you for spotting this.

**Figure 1:** The blue vectors along the West Spitsbergen Shelf are difficult to distinguish—consider making them thinner. Also, please bold the abbreviations for improved readability, particularly for colorblind readers.

⇒ We have added boxes around the labels to increase the readability and adjusted the arrow thickness as suggested.

**Line 69:** Is it possible to label the STC on the map in Figure 1? It seems to correspond to vectors branching onshore from the WSC, but this wasn't immediately clear until Section 3.2.

⇒ We have added a label and try to connect it with the respective arrows on the map using a black connection line (as suggested by the other review addressed in the same process)

**Line 137:** The phrase "expansion of multiple and relatively warmer isotherms... at several latitudes" is vague. Please consider quantifying this statement. Relevant details in Appendix B should be moved to the methods section.

⇒ We have moved the explanations on how we identify a warming event to Section 2.2 in the main body of the manuscript and rewritten the sentence in question to be more specific.

**Line 149:** Current properties are in the appendix too, please move relevant details to the methods section.

⇒ Yes, we have moved this into Section 2.2 in the main body of the manuscript as well.

**Line 154:** This equation is important and should be included in the manuscript body.

⇒ The equations for the calculation of the drag coefficient as well as the subsequent calculation of the wind stress and the Ekman transport have been included in the manuscript.

**Line 168:** Please include the standard deviation alongside the mean value.

⇒ Done.

**Line 191:** Can you quantify the tilt mentioned here?

⇒ We have added values for this into the text.

**Figure 7:** As with Line 149, it's unclear how the "core" is defined. This information should be relocated from the appendix to the methods. Also, subplots (b) and (c) are on log scales—please indicate this in the caption.

⇒ The caption has been updated to indicate the log scales of the two subpanels. The information on how current branch and core position were defined has been moved into Section 2.2 in the main body of the manuscript.

**Technical Comments**

**Line 21:** The sentence on the cold vs warm pathways is a monster. As someone not intimately familiar with this region, I needed to go back and forth between the text and Figure 1 a lot. It would be helpful for the reader to break this up into 2 sentences. Particularly, consider breaking it where you go into further details on where the West Spitsbergen Current originates/divides.

⟹ Very valid feedback. We have improved the readability of this part by dividing it into smaller parts and shorter sentences.

**Line 165:** Did you mean to write "actual"? Please revise for clarity.

⟹ This part has been re-written, and we hope the content is clearer now.

**Lines 187 & 197:** "Subside" might be better replaced with "subduct," depending on the intended physical process.

⟹ Yes, this is exactly what we mean. Great to get such detailed input from somebody with English as mother tongue, thank you.

**Line 277:** This sentence is overly complex, largely due to excessive parenthetical phrasing. A revision is recommended, and incorporating the appendix details into the main text would reduce the need for such parentheses.

⟹ This part has been rewritten and simplified, and the relevant details on the current identification can now be found in Section 2.2 in the main body of the manuscript.

**Line 328:** "Increase" should be "increases" to maintain subject-verb agreement.

⟹ Again, thank you for such detailed feedback on the grammar.

**Figure 9:** It would make more sense for all lines representing slope water to be colored the same and solid, and all lines representing shelf waters to be colored the same and dashed. Label the panels after naming them in the caption.

⟹ Thank you for your valuable suggestions. We have revised this figure accordingly and also added a small legend for improved readability.

---

## Author Comment (AC3)

**Mechanisms of Warm-Water Intrusions onto the West Spitsbergen Shelf during Winter**

Lukas Frank[1,2], Jon Albretsen[3], Ragnheid Skogseth[1], Frank Nilsen[1,2], and Marius O. Jonassen[1,2]

[1]The University Centre in Svalbard, Longyearbyen, Norway
[2]Geophysical Institute, University of Bergen, Bergen, Norway
[3]Institute of Marine Research, Bergen, Norway

[revised manuscript text omitted]

**Appendix A:  Frequency Filtering**

With a temporal resolution of 1 day, the ROMS model data time series constitute of variability over a wide frequency range, from short-term inertial oscillations to the annual cycle. Since the major warming events of interest in this study occur over characteristic time scales of several days to weeks, it is necessary to separate the corresponding low-frequency variability from higher-frequency fluctuations, particularly those caused by transient eddies, internal waves and direct wind forcing close to the surface. To achieve this, we apply a fourth-order Butterworth low-pass filter with a cutoff frequency of 20 days to the ROMS hydrographic and current time series. Thanks to its flat frequency response in the passband, this filter ensures minimal distortion of the retained signal. The effects of this filtering can be seen in Figure A1, where the smoothed time series highlight the longer-term variability of interest while suppressing higher-frequency fluctuations.

**Appendix B:  Identification of Warming Events**

As outlined in Section 2.2 of this study, we used a pseudo-Lagrangian approach to identify warming events on the WSS. In contrast to the classical Eulerian view on the data (analyzing e.g. the temperature time series at a fixed point in time, such as a mooring location), this allows for more variability in the extent/strength of what is identified as a warming event.

In order to account for a wide range of initial/boundary conditions during a specific warming event (e.g. the WSC core temperature at the time or the temperature of the water residing on the WSS prior to the warming), 0.2°C-increment-isotherms where chosen covering the whole temperature range of the data set (-1.6°C − 8.2°C). In case of multiple, separate contour elements corresponding to the same isothermal present at the same section at the same time step, the largest one originating over the slope (westernmost point of the contour at negative x) and penetrating onto the shelf (easternmost point of the contour at positive x) was chosen for the further processing steps. For all contour elements, the position of their easternmost point was tracked independently at each latitude section throughout the study period (see example shown in Figure 3 (a)). A warming event could then be identified as an increase in the x-locations of the tracked positions (Figure 3 (b)) based on multiple contour levels. Smaller events (based on only a few contour levels, not detected at an extended latitude-range etc.) were manually discarded for the further analysis. In the end, the heat content increase at any position on the shelf was calculated as the temperature difference between the isotherms located at that position before and after the warming, multiplied by the mean density (1028 ) and the specific heat capacity of sea water (3985 ). (a) Temperature cross section at 77.6° N on 01.01.2021. The black lines indicate the isotherms used to identify the warming events. The 3°C-isothermal is highlighted in green as an example and its easternmost point is marked with a round marker. (b) Time series of the on-shelf x-position of the easternmost points (the green marker for the example contour in (a)) for all isotherms during the winter season 2020/21. Those periods identified as a warming event are marked with color, corresponding to the value of the respective isotherm. The black arrow indicates the time of the cross section snapshot shown in (a).

**Appendix B:  Identification of Current Branches**

[Figure]

**Figure A1.** Time series of (a) temperature, (b) salinity and (c) meridional current speed at an example point from the WSS break section at 77.9° N at x=5 km in 10 m depth. The light curves in the background display the raw daily values, the opaque curves on top show the low-pass filtered smooth time series used for the further analyses.

520 As stated in the corresponding Section 2.2 of the main manuscript, the core positions of individual current branches were identified as local maxima in the northward current cross-section at each latitude and time step. An example cross-section is given in Figure 4. Minor peaks with a maximum northward speed of less than 0.05 or a prominence of less than 20% of their maximum speed were discarded (the grey markers in Figure 4). Based on contour lines with a spacing of 0.02 m/s, the lowest contour solely contributing to the remaining peaks, peak were selected (the colored contours in Figure 4 (a)). In the end, 
[revised manuscript text omitted]